# A Review of the Impact of Mycotoxins on Dairy Cattle Health: Challenges for Food Safety and Dairy Production in Sub-Saharan Africa

**DOI:** 10.3390/toxins12040222

**Published:** 2020-04-02

**Authors:** David Chebutia Kemboi, Gunther Antonissen, Phillis E. Ochieng, Siska Croubels, Sheila Okoth, Erastus K. Kangethe, Johannes Faas, Johanna F. Lindahl, James K. Gathumbi

**Affiliations:** 1Department of Pathology, Parasitology and Microbiology, Faculty of Veterinary Medicine, University of Nairobi, PO Box 29053, 00100 Nairobi, Kenya; kemboidc@gmail.com; 2Department of Animal Science, Chuka University, P.O Box 109-00625 Chuka, Kenya; 3Department of Pharmacology, Toxicology and Biochemistry, Faculty of Veterinary Medicine, Ghent University, Salisburylaan 133, 9820 Merelbeke, Belgium; Gunther.Antonissen@ugent.be (G.A.); phillisemelda.ochieng@ugent.be (P.E.O.); Siska.Croubels@ugent.be (S.C.); 4Department of Pathology, Bacteriology and Avian Diseases, Faculty of Veterinary Medicine, Ghent University, Salisburylaan 133, 9820 Merelbeke, Belgium; 5Department of Food Sciences, University of Liège, Faculty of Veterinary Medicine, Avenue de Cureghem 10, 4000 Liège, Belgium; 6School of Biological Sciences, University of Nairobi, P.O Box 30197-00100 Nairobi, Kenya; dorisokoth@yahoo.com; 7P.O Box 34405, 00100 Nairobi, Kenya; mburiajudith@gmail.com; 8BIOMIN Research Center, Technopark 1, 3430 Tulln, Austria; johannes.faas@biomin.net; 9Department of Biosciences, International Livestock Research Institute (ILRI), P.O Box 30709, 00100 Nairobi, Kenya; 10Department of Medical Biochemistry and Microbiology, Uppsala University, P.O Box 582, 751 23 Uppsala, Sweden; 11Department of Clinical Sciences, Swedish University of Agricultural Sciences, P.O Box 7054, 750 07 Uppsala, Sweden

**Keywords:** mycotoxins, dairy, aflatoxin, Sub-Saharan Africa, aflatoxin M1

## Abstract

Mycotoxins are secondary metabolites of fungi that contaminate food and feed and have a significant negative impact on human and animal health and productivity. The tropical condition in Sub-Saharan Africa (SSA) together with poor storage of feed promotes fungal growth and subsequent mycotoxin production. Aflatoxins (AF) produced by *Aspergillus* species, fumonisins (FUM), zearalenone (ZEN), T-2 toxin (T-2), and deoxynivalenol (DON) produced by *Fusarium* species, and ochratoxin A (OTA) produced by *Penicillium* and *Aspergillus* species are well-known mycotoxins of agricultural importance. Consumption of feed contaminated with these toxins may cause mycotoxicoses in animals, characterized by a range of clinical signs depending on the toxin, and losses in the animal industry. In SSA, contamination of dairy feed with mycotoxins has been frequently reported, which poses a serious constraint to animal health and productivity, and is also a hazard to human health since some mycotoxins and their metabolites are excreted in milk, especially aflatoxin M1. This review describes the major mycotoxins, their occurrence, and impact in dairy cattle diets in SSA highlighting the problems related to animal health, productivity, and food safety and the up-to-date post-harvest mitigation strategies for the prevention and reduction of contamination of dairy feed.

## 1. Introduction

The livestock sector accounts for about 18% of the gross domestic product (GDP) in Sub-Saharan Africa (SSA). In 2017, there were 35.3 million tons of milk produced by 71.1 million dairy cattle in Africa, of which 66.5% is located in SSA [1]. However, Africa is producing only 5.1% of the world milk and the total yield per cow per year is also low [1]. With a population of about 1 billion people, projected to rise to over 1.2 billion by 2025 [1], there is an urgent need for improved productivity per animal since the increased production of milk over the years was rather the result of an increased number of animals than increased individual animal productivity. This may be attributed to the traditional (pastoralism) system that is mostly practiced in SSA. However, due to human population growth, increased milk consumption per capita, and land shortage and increasing interest in production, semi-intensive and intensive dairy farming systems are increasingly being adopted [2]. In these semi-intensive and intensive systems, mostly localized in rural and peri-urban regions, the animals either graze or are fed on planted fodder or crop residues supplemented with concentrates [3]. Cereal grains are the major ingredients of most of the concentrates and these are often of substandard grade, mostly due to fungal growth and discoloration and thus considered unfit for human consumption, which predisposes these feeds to mycotoxin contamination [4]. 

Mycotoxins are secondary metabolites of fungi that contaminate food and feed and have a significant negative impact on human and animal health including animal productivity. *Aspergillus, Fusarium,* and *Penicillium* are the major mycotoxin-producing fungi. These toxigenic fungi are classified into two groups; field fungi that contaminate crops and produce toxins while still on the field such as *Fusarium* species, and storage fungi, such as *Aspergillus* and *Penicillium* species, that mainly produce toxins after harvesting during storage. Production of mycotoxins is related to environmental conditions, stress to the plant, and damage to the grains caused by rodents and pests and abiotic factors such as pH of feed and moisture content [5,6]. The tropical condition in SSA together with poor storage of feed promotes fungal growth and subsequent mycotoxin production [6,7]. 

Aflatoxins (AF) produced by *Aspergillus* species, fumonisins (FUM), zearalenone (ZEN), T-2 toxin (T-2), and deoxynivalenol (DON) produced by *Fusarium* species, and ochratoxin A (OTA) produced by *Aspergillus* and *Penicillium* species are well-known mycotoxins of major agricultural importance and have been found concurrently occurring in feeds, with AF, which is a class 1 carcinogen to human beings, the most prevalent and also the most studied in SSA [8,9,10,11]. 

Consumption of feed contaminated with these toxins may cause mycotoxicoses in animals, characterized by a range of clinical signs depending on the toxin, and may cause losses in the animal industry. In SSA, contamination of dairy feed with mycotoxins has been frequently reported, which poses a serious constraint to animal health and productivity and is also a hazard to human health. This is because some mycotoxins and their metabolites are excreted in milk, especially aflatoxin M1 (AFM1). This review describes the major mycotoxins and their occurrence and impact in dairy cattle diets in SSA, highlighting the problems related to animal health and productivity, food safety, and provides the up-to-date post-harvest mitigation strategies for the prevention and reduction of contamination of dairy feed.

## 2. Legislation of Mycotoxins in Africa

As per 2003, 15 countries in Africa had mycotoxin regulations covering 59% of the inhabitants of Africa. These regulations only are concerned with AF except for South Africa where guidance levels exist for ZEN, FUM, and DON in dairy feeds [10,12]. Regionally, East Africa Community (EAC) harmonized and set regulatory limits for AFB1 at 5 µg/kg in dairy feed and 0.5 µg/kg for AFM1 in milk. In addition, Rwanda through the Rwanda Standards Board (RSB) has established a regulatory limit of 5 µg/kg for AFB1 in cattle feed supplements [13]. No other regional regulatory limits have been established and implemented in Africa. In West Africa, Nigeria through the National Agency for Food and Drug Administration and Control has set a regulatory limit of 5 µg/kg and 0.5 µg/kg for AFB1 in dairy feed and AFM1 in milk, respectively [14]. Cote d’ Ivoire has set a limit of 10 μg/kg for total AF in complete feed, while Senegal has a limit of 50 μg/kg for AFB1 for animal feeds from peanuts [12]. In Southern Africa, Republic of South Africa has a regulatory limit for AF and guidance levels for other mycotoxins in dairy feed, i.e., at 5 μg/kg for AFB1, 50,000 μg/kg for FUM, 500 μg/kg for ZEN, and 3000 μg/kg for DON. The regulatory limit for AFM1 in milk is set at 0.05 μg/kg [10]. Mozambique has set regulatory limits for total AF at 10 μg/kg in dairy feed, while Zimbabwe has not set the limit for dairy feed but use a 10 μg/kg limit for poultry feed [12]. Worldwide, the World Health Organization/Food and Agriculture Organization of the United Nations (WHO/FAO) through the Codex Alimentarius Commission (CODEX) have set up a regulatory limit for AFB1 in dairy feeds at 5 µg/kg and for AFM1 in milk at 0.5 µg/kg [15]. The European Union (EU) and the United States of America through the United States Food and Drug Agency (USFDA) have also established a regulatory limit for AF and guidance limits for other mycotoxins, as shown in Table 1.

Sirma et al. [16] noted that countries with mycotoxin problems including most SSA countries tend to have laxer enforcement of the regulations set and this may be due to the countries setting limits that are beyond their capacity to implement. Some developed regions such as the EU, USA, and Canada have set up regulations that allow the contaminated feed to be used in less susceptible species. The EAC through Policy Brief on Aflatoxin Prevention and Control (Policy Brief No. 8, 2018) has recommended setting up a policy for cascading for direct utilization of AF-contaminated food based on the level of contamination and negative effects to the animal species, however, this has not been established yet in EAC and other SSA countries [16,17]. Local influencing factors such as enforcement capacity, levels of contamination with mycotoxins, and existing regional standards and the use of these commodities need to be considered in order to improve acceptability and uptake of these regulations [16].

## 3. Incidence of Mycotoxins in Dairy Feed in Sub-Saharan Africa

In SSA, dairy cattle are fed on fodder that includes grazed grass, hay, legumes, and silage and supplemented with concentrates. These concentrates are often compounded feeds, grain millings, and oilseed cakes. The compounded feed is made from mixing raw materials such as cereals including maize, small grains, oil cakes such as cotton seed cake, sunflower cake, soy meal cake, copra, noug seed, and fish meal [10,20,21]. Taking into account the feed and food shortage in SSA, spoiled and moldy maize that has been considered unfit for human consumption is often fed to animals [22,23]. Both fodder and concentrates have been reported to be contaminated with mycotoxins, with the latter being the major source of contamination [11,24,25,26]. This makes all dairy farming practiced in SSA at risk of contamination with mycotoxins. Table 2 is a summary of reported cases of dairy feed contamination in SSA, most of which are above the maximum regulatory levels set by national institutions, regional bodies, and the European Union (EU). Overall, aflatoxins (AFs) are the most commonly tested and detected mycotoxins in both finished feed and raw material with a maximum level of 9661 µg/kg [27]. Fumonisins (FUMs) were the second most common mycotoxins with sorted bad maize used for animal feed from Tanzania having the highest mean level of 14 mg/kg [28]. OTA was reported in South Africa, Kenya, Nigeria, and Sudan with the highest level of 19 µg/kg in Sudan. However, the sample sizes in these countries were too small for a proper comparison between countries. The means of the positive samples were between 2 and 15 µg/kg [29] and below the 50 µg/kg EU guidance limit for OTA. The occurrence of OTA seems to follow a similar pattern to AF in areas where AF commonly occur and this may be because they are both produced by *Aspergillus* species. HT-2 toxin was only reported in South Africa [11].

AF and FUM have been widely studied in SSA due to their frequent occurrence in food and feed [29] and high toxicity to animals and humans. This also makes them be regulated in most countries [10]. However, recently, other mycotoxins have been reported in dairy feed in South Africa [4,10,11], Kenya [6,21,30], Rwanda [13], Tanzania, Sudan, Ghana, and Nigeria [21]. Furthermore, raw materials used for compounded feed preparation can simultaneously be contaminated by different mycotoxins since some mycotoxigenic fungi grow and produce mycotoxins under similar conditions [31]. Synergistic, additive, and antagonistic effects due to co-occurrence of mycotoxins occur with, for example, FUM, reported to increase the uptake of AF and subsequently the carry-over to milk [32].

### 3.1. East Africa

Mycotoxins were reported in compounded dairy feeds, forages, and raw materials used for making compounded dairy feeds. Countries from East Africa that are near the equator have a higher occurrence of AF than other mycotoxins due to the warm and humid climate, which promotes the growth of *Aspergillus* species.

In Ethiopia, Gizachew et al. [33] reported a 100% incidence of AFB1 in compounded dairy feed, brewer yeast, silage, maize, and pea hull with all of the samples above the EU and EAC regulatory limit of 5 µg/kg. Similarly, in Kenya, Okoth and Kola [34] reported a 100% incidence of AF in compounded dairy feed, cottonseed cake, and sunflower seed cake at levels ranging from 5.13 to 1123 µg/kg. In the study, 95% of the samples were above the regulatory limit of 10 µg/kg with cottonseed cake and compounded dairy feed having a higher number of samples above the regulatory limit at 51.2% and 41.9%, respectively; 7% of the sunflower seed cake samples exceeded the regulatory limit. However, in Tanzania, Mohammed et al. [36] reported 61.5% of sunflower-based dairy feed having AFB1 above the EAC and EU regulatory limit and an overall incidence of 65%. Senerwa et al. [27] in Kenya reported a higher level of samples having AFB1 above the EAC and EU regulatory limit in compounded dairy feed at 90.3% but noted differences between different agro-ecological zones. Moldy maize used as animal feed in Kenya [23] and Tanzania [28] had AF and AFB1, respectively, at 56% and 29%; however, the means (3.84 µg/kg and 3.49 µg/kg respectively) were below the EAC and EU regulatory limit. In Sudan, Rodrigues et al. [21] reported an incidence of 54% of AF in compounded dairy feed and raw material used for making compounded dairy feed.

FUM are the second most common mycotoxins reported. Rodrigues et al. [21] reported 76% incidence of FUM in compounded animal feed, grains, and other feed commodities in Kenya with a mean of 956 µg/kg, which is below the EU guidance limit for FUM of 50,000 µg/kg. In Tanzania, Nyangi et al. [28] reported incidence of 60% and 51% of FUM in spoilt maize unfit for human consumption sorted from good maize after harvesting and meant for animal feed and maize bran, with the spoilt maize having a higher mean of 14,450 µg/kg, which is still below the EU guidance limit. Type B trichothecenes (DON, 3/15-acetyl-deoxynivalenol and nivalenol) were also reported in Kenya [6,21] and Sudan [21] at incidences of between 33% and 63% in compounded dairy feed, raw materials for animal feed, and forages. ZEN and OTA were also reported in dairy feed in Kenya and Sudan; however, few numbers of samples were analyzed for OTA for comparison [21].

### 3.2. Central Africa

Little information concerning mycotoxins contamination in feed is available for Central Africa. This is attributed to the lack of knowledge on the mycotoxin issue, poverty, and lack of research facilities, manpower, and skills in these countries [37]. However, in Burundi and the Democratic Republic of Congo, AFB1 has been reported in food samples and AFM1 in cow milk [38], but little data is available on mycotoxins contamination in dairy feed. Raphaël et al. [39] reported the occurrence of AF in poultry feed, maize, and peanut meal at levels as high as 950 µg/kg in Cameroon, but little data is available on dairy feed.

### 3.3. West Africa

AF and FUM were the most prevalent mycotoxins in Ghana (72% and 94%, respectively) and Nigeria (100% and 89%, respectively) [21,35]. Rodrigues et al. [21] reported the highest level of AF in compounded dairy feed and raw materials for making compounded feed in Nigeria at 435.9 µg/kg and a mean of positive of 115 µg/kg [21]. However, another study by Omeiza et al. [35] on compounded dairy feed and pasture showed a lower mean of 10.5 µg/kg, with 66.4% of the samples exceeding the EU regulatory limit. In the study, concentrates had the highest incidence of AFB1 at 93% with dry pasture having 60% and pasture mixed with concentrates having an incidence of 86.9%, indicating concentrates as the major source of AF. In Ghana, the highest level was 199 µg/kg with a mean of 26 µg/kg in compounded dairy feed and raw materials [21]. The highest level and mean of FUM reported in the region were in Nigeria (2860 µg/kg and 919 µg/kg, respectively) [21]. Other mycotoxins reported are ZEN, OTA, and type B trichothecenes (DON, 3/15-acetyl-deoxynivalenol, and nivalenol). This occurrence pattern of mycotoxins is similar to that of East Africa and this may be because countries in West Africa and East Africa are both near the equator, hence similar environmental conditions such as the higher levels of humidity.

### 3.4. South Africa

South Africa exhibited higher contamination of *Fusarium* mycotoxins FUM, DON, and ZEN [21]. This is in line with the findings by Gruber-Dorninger et al. [29] who reported DON, FUM, and ZEN as the most common mycotoxins in South Africa. FUM had an incidence of between 57% and 100%. The highest level of FUM was reported by Rodrigues et al. [21] at 4.4 mg/kg in commercial dairy feed, grain, and other feed commodities, with the highest mean level reported by Njobeh et al. [10] in compounded dairy feed being 0.98 mg/kg and below the EU guidance limit and South African limit of 50 mg/kg. Changwa et al. [11] and Njobeh et al. [10] reported similar occurrence patterns and incidences of DON and ZEN in dairy feed. This is also in line with the findings of Gruber-Dorninger et al. [29] where DON and ZEN are commonly reported to occur together since they are produced by the same species of fungi, but DON occurring at higher levels. Njobeh et al. [10] reported an incidence of 96% of both DON and ZEN in compounded feed and Changwa et al. [11] an incidence of 60% of both DON and ZEN in assorted dairy feeds. In both studies, compounded dairy feed had the highest level of DON. Rodrigues et al. [21] reported an incidence of 87% for type B trichothecenes (DON, 3/15-acetyl-deoxynivalenol, and nivalenol) and a lower incidence of 29% for ZEN. The highest reported level of DON (2280 µg/kg) [10] and ZEN (195 µg/kg) [21] were below the EU guidance limit for DON (5000 µg/kg) and ZEN (500 µg/kg), respectively, and the South African limit 3000 µg/kg and 500 µg/kg, respectively.

AF occurred at 6%–100%, with aflatoxin G2 (AFG2) being the most prevalent in compounded dairy feed and raw materials for animal feed. Changwa et al. [11] reported AFG2 and AFB2 as the most frequent AFs, with AFG2 having the highest mean concentration of 41 µg/kg in maize silage, grass, total mixed ratio, brewer yeast, molasses, and maize bran. AFB1 was the least frequent AF, with the highest level reported in lucerne based feeds (mean 2.1 µg/kg). Incidence and levels of AFG2 in the study were high, 62% of the positive samples exceeded the Food and Drug Agency (FDA) regulatory limit for AF of 20 µg/kg. However, Njobeh et al. [10] and Rodrigues et al. [21] reported lower levels of samples exceeding the regulatory limit (mean 14.7 µg/kg) in compounded dairy feed and 0.2 µg/kg in compounded feed and raw materials, respectively). Other mycotoxins reported were OTA and HT-2 toxin. Njobeh et al. [10] reported OTA at 16%, but Rodrigues et al. [21] and Changwa et al. [11] did not detect OTA. Changwa et al. [11] reported 87.5% incidence of HT-2 toxin with Njobeh et al. [10] and Rodrigues et al. [21] not detecting T-2 and HT-2 in compounded dairy feed and raw materials.

## 4. Impact of Mycotoxins in Dairy

Mycotoxins in Africa have an impact on food security and safety, animal and human health, international trade, and national budgets, leading to reduced self-sustainability and increased reliance on foreign aid [37]. In the dairy sector, contamination of feed with mycotoxins causes serious economic and food security and safety issues. Economic impact occurs through the direct market costs associated with lost trade or reduced revenues due to the rejection of contaminated animal products and reduced productivity, death of the animal especially calves which are more sensitive, and increased cost of treatment and mycotoxin mitigation [37]. Some mycotoxins, including AF and T-2, are immunosuppressive in cattle, leading to vaccination failure and increased susceptibility to infectious diseases [40] with hidden cost affecting animal health and productivity. The impact is borne by all participants in the dairy sector including feed manufacturers, dairy farmers, milk processors, and consumers [21]. Little has been done to financially quantify the cost of mycotoxins exposure in the dairy enterprise in Africa. In Kenya, Senerwa et al. [27] reported 61.4% of feed contaminated with AFB1 above the FAO/WHO/Kenya limit of 5 µg/kg. This translates to a possible economic cost per year for dairy feed manufacturers of 22.2 billion US $ and, additionally, a further 37.4 million US $ is incurred in losses by farmers annually due to reduced milk yield as a result of feeding cattle with feed contaminated with AFB1 [27]. In the same study, 10.3% of the milk samples had AF levels above the WHO/FAO limit of 0.5 µg/kg, which would cost dairy farmers 113.4 million US $ per year if legislation was enforced.

Animal health issues due to mycotoxicoses affect both the health and productivity of animals with the clinical signs manifested depending on the individual mycotoxin as shown in Table 3. There are two forms of mycotoxicoses, acute mycotoxicoses that occur due to consumption of a high single dose of mycotoxins and chronic mycotoxicoses due to chronic consumption of low levels of mycotoxins over time. Recorded toxic levels of mycotoxins that cause acute disease in dairy cattle are 100 µg/kg for AF, 400 µg/kg for ZEN, and above 100 µg/kg for T-2 [41]; however, chronic aflatoxicosis caused by low-level exposure of mycotoxins over time poses a more common health problem to the animals and also food safety concern to humans. Generally, mycotoxins cause reduced feed intake, alter ruminal fermentation and reduce feed utilization, suppressing immunity, alter reproduction, and cause hepatotoxicity and nephrotoxicity [42]. In comparison, ruminants can be less severely affected by certain mycotoxins compared to monogastrics, which is attributed to the microbial activity in the rumen that can modify the mycotoxin chemical structure into less toxic compounds. Upandaya et al. [43] conducted an in vitro study on the degradation of AFB1 by ruminal fluid from Holstein cattle using 80 µg/kg AFB1 and reported a degradation starting after 3 h incubation with an eventual 14% reduction of AFB1 by 12 h. In agreement, Jiang et al. [44] also performed an in vitro study with rumen fluid collected from Holsteins fed with two substrate alfalfa hay (HA) and ryegrass hay (HR) and, after 72 h incubation, there was a decrease of 83% for HA and 84% for HR of included AFB1 (960 ng/mL). However, in both studies, the metabolites formed due to the degradation were not reported.

Fumonisins are minimally absorbed by ruminants with most of it being excreted in the unmetabolized form in feces. Gurung et al. [45] in an in vitro study using 100 mg/kg FB1/kg reported minimal degradation of FB1 (10%) to hydroxylized FB1 (HFB1) by ruminal microbiota after incubation for 72 h. Similarly, goats fed diets containing 95 mg FB1/kg for 112 days excreted 50% of the FB1 in the unmetabolized form in feces [46].

The intact ruminal epithelium is an effective barrier against DON and ZEN [47]. DON is degraded by the ruminal microbiota to the less toxic metabolite de-epoxy DON (DOM-1) [48,49]. A study by Seeling et al. [48] reported 94%–99% biotransformation of DON to DOM-1. Keese et al. [47] reported no significant amount of unmetabolized DON passing through the ruminal epithelium in cattle fed 50% concentrate proportion and 5.3 mg DON/kg DM, nor if a ration with 60% concentrate and 4.6 mg/kg DM fed for a total period of 29 weeks; however, DOM-1 was present in serum, which indicates systemic uptake. Valgaeren et al. [49] studied the role of roughage provision on the absorption and disposition of DON and its acetylated derivatives in calves and observed an absolute DON oral bioavailability of 4.1% in ruminating calves compared to 50.7% in non-ruminating calves, indicating the ability of the rumen to degrade DON. Recently, Debevere et al. [50] showed the degradation of DON was hampered when ruminal pH was low (pH 5.8), as is the case in cattle suffering from sub-acute rumen acidosis. Interestingly, when rumen inoculum of non-lactating cows was used, a similarly reduced degradation of DON was seen.

An in vitro study by Kiessling et al. [51] on the effect of ruminal microbiota on OTA, ZEN, and T-2 after incubation between 30 min to 3 h showed degradation of OTA to ochratoxin α and phenylalanine, which are non-toxic metabolites mainly produced by the action of protozoa. This ruminal fauna of protozoa can also be affected by diet and the variation in the ruminal protozoa population will affect mycotoxin degradation. Kiessling et al. [51] reported a decrease of 20% of OTA degradation in sheep fed with a high hay concentrate ratio (3:7 weight/weight) compared to a low hay concentrate ratio (5:4 weight/weight). An in vivo study in sheep with 5 mg/kg OTA showed no OTA in blood [51]. For ZEN, the conversion was to α-zearalenol and to a lesser extent to β-zearalenol. α-zearalenol is more toxic and more easily absorbed from the intestines into the bloodstream as compared to the parent compound due to increased polarity, indicating the action of the ruminal microbiota can also increase the toxicity of ZEN. Kiessling et al. [51] also reported in vitro conversion of T-2 to HT-2 toxin after 30 min of incubation with ruminal fluid collected from sheep.

### 4.1. Aflatoxins

Field and experimental aflatoxicosis have been previously described in dairy cattle. Van Halderen et al. [52] reported a field outbreak mortality of 7 out of 25 calves in South Africa fed rations containing locally produced maize with 11,790 µg/kg AF. Clinical signs included loss in body mass, rough hair coat, diarrhea, and rectal prolapse. Outside SSA, Mckenzie et al. [53] reported acute aflatoxicosis that was believed to have caused mortality of 12 to 90 drought-stricken calves in Australia. The calves were fed peanut hay that was later analyzed and determined to contain 2230 µg/kg AF. More recently, Umar et al. [54] reported 45 field cases of aflatoxicosis on a local farm in Okara (Punjab, Pakistan). The cows were fed corn-rich forage with 33,500 µg/kg AF. The clinical signs were anorexia, depression, photosensitization, and diarrhea, with 15 animals dying.

Experimental studies have described aflatoxicosis, with clinical signs being reduced feed intake and feed conversion, reduced milk production, reduced reproduction capacity, lameness, immunosuppression, hepatotoxicity, and nephrotoxicity [41,42]. Applebaum et al. [55] reported a significant decrease in milk production in cattle fed with 13 mg AFB1 per day for 7 days. Likewise, Ogunade et al. [56] and Jiang et al. [57] also reported a numerical drop in milk yield in cows fed 75 µg/kg Dry Matter Intake (DMI) (1725 µg/head per day) for 5 days. Sulzberger et al. [58] reported depression in milk yield and feed conversion at 100 µg AFB1/kg of DMI. In contrary, studies by Queiroz et al. [59] on cattle fed 75 µg/kg DMI (1725 µg/head per day) for 4 days, Kutz et al. [60] feeding 112 μg of AFB1/kg of Total Mixed Ration (TMR) DMI to dairy cows in early to mid-lactation for 7 days, Sumantri et al. [61] on cattle fed 350 μg AFB1/cow/day for 10 days, and Mosoero et al. [62] using lower levels of AF of 0.16 µg/kg BW and 3.41 μg AFB1/cow/day for 3 days showed no health effect [62]. Exposure to AF also affects rumen fermentation, reducing the utilization of nutrients, and eventually may affect animal productivity. Mesgaran et al. [63] reported reduced gas production, dry matter digestibility, and ammonia-N concentrations caused by AFB1 in vitro. Jiang et al. [44] also reported similar results with AFB1 affecting in vitro fermentation characteristics in terms of reduced ammonia N and volatile fatty acids (VFA) concentrations but without reducing dry matter digestibility or affecting VFA pattern. Based on the levels of AF reported in dairy feed in SSA, chronic aflatoxicosis is a risk in the dairy sector.

### 4.2. Deoxynivalenol

DON, also called vomitoxin, induces anorexia and emesis in humans and animals. This is usually achieved by affecting the chemoreceptor trigger centers and causing gastrointestinal lesions. Pigs are the most sensitive species while cattle are less susceptible. DON affects ruminal fermentation and causes reduced milk yield [41]. Contradictory results have been shown on the effect of DON on feed intake. Trenholm et al. [64] studied the effect of a diet contaminated with DON in non-lactating Holstein dairy cows fed at a dose of 1.5 mg/kg and 6.4 mg/kg of feed for 6 weeks, there was no adverse effect observed, however, there was a slight decline in feed consumption following the change from the low DON dose (1.5 mg/kg) to the high DON dose (6.4 mg/kg). In contrary, Seeling et al. [41] reported no effect on feed intake on feeding approximately 3.4 mg DON per kg at a reference DM of 88% complete ration, similar to Winkler et al. using 5 mg DON/kg feed showed no effect on performance parameters on dairy cattle [65]. In calves, Valgaeren et al. [49] reported severe liver failure in 2-3 months old calves with no functioning rumen induced by 1.13 mg DON/kg feed, indicating the significance of rumen microbiota in DON degradation.

### 4.3. Fumonisins

Ruminants are more resistant to FUM toxicity than horses and pigs [66]. Field outbreaks have been reported in horses and pigs but not in dairy cattle. Fumonisins act by altering sphingolipid biosynthesis hence leading to the accumulation of sphinganine and causing toxicity. Oral administration to calves with a diet containing FB1 at 2.36 mg/kg/day increased to 3.54 mg/kg/day for 239 to 253 days showed elevated sphinganine/sphingosine ratios with mild hepatocellular morphology changes accompanied by mild bile duct epithelial changes [67]. Feeding trials with 75 mg/kg, 94 mg/kg, and 105 mg/kg FB1 for 14 days, 253 days, and 31 days, respectively, have also been reported to cause reduced milk yield, reduced feed intake, hepatotoxicity, nephrotoxicity, and reproduction problems and, hence, it can be concluded that oral administration with levels above 75 mg/kg is toxic to cattle [41,42,68]. Experimental administration of 1 mg/kg of FB1 intravenously to calves for 7 days caused lethargy, loss of appetite, hepatotoxicity, and nephrotoxicity [66].

### 4.4. Ochratoxin A

Ochratoxicosis is rarely reported in cattle. This is attributed to the ability of the rumen microbiota to easily degrade OTA to non-toxic forms as demonstrated by Kiessling et al. [51]. Ribelin et al. [69] reported anorexia, diarrhea, difficulty in rising and cessation of milk production with recovery on the 4th day in cattle fed a high single dose of OTA of 13.3 mg/kg; this dose can be rarely achieved in the field and low doses of 0.2 mg/kg, 0.75 mg/kg, and 1.66 mg/kg for 5 days produced no clinical disease. With the highest level of OTA reported in SSA being 19 µg/kg, which is way lower than those used in the experiment, it can, therefore, be concluded that OTA is rarely a problem in dairy cattle in SSA.

### 4.5. T-2 toxin

In dairy cattle, T-2 has been associated with hemorrhagic gastroenteritis [41], feed refusal, and gastrointestinal lesions [70]. Weaver et al. [70] reported severe depression, hindquarter ataxia, knuckling of the rear feet, listlessness, and anorexia in a calf fed 0.6 mg T-2/kg body weight for seven consecutive days. Reduction in milk yield and the absence of estrus have also been associated with T-2 [41].

### 4.6. Zearalenone

Zearalenone has an estrogenic response in cattle causing abortion and changes in the reproductive organs. A case report by Kallela and Ettala [71] reported early abortion in cattle feeding on hay containing 10 mg/kg ZEN. Abnormal estrus cycle, vaginitis, behavioral estrus in pregnant animals, mammary development in pre-puberty heifers, and sterility have also been reported in cattle fed with feed containing 1.5 mg ZEN/kg feed [72]. Experimental studies using 500 mg and 250 mg of 99% purified ZEN in a gelatin capsule orally in lactating dairy cattle and virgin heifers, respectively, showed no effects except for depression in the conception rate in the virgin heifers [73,74].

## 5. Food Safety and Hazards of Mycotoxins

Besides the effects on animal health, some mycotoxins can pass to milk causing food safety issues and posing a hazard to human health. Of all the studied mycotoxins, only AF has been described to be transferred to the milk of lactating cattle in significant levels of concern. This is of great importance to public health worldwide as the toxin is classified as a carcinogen and infants, the primary consumers of milk, are more susceptible. Negligible transfer of FUM, ZEN, OTA, and DON has been reported, but the health impacts of this are unknown. Carry-over studies and surveys of mycotoxins in milk in SSA have not been extensively carried out, except for surveys of AFM1.

Once ingested by ruminants, part of the ingested AFB1 is degraded in the rumen. Kiessling et al. [51] suggested that the type of microbiota in the rumen will determine the level of degradation and is dependent on species, age, sex, and breed. Upadhaya et al. [43] further reported AFB1 degradation of 14% in cattle compared to 25% in goats with type of feed also determining the level of degradation. Similarly, Jiang et al. [44] reported a higher degradation of AFB1 that was dependent on the type of feed. The type of feed has an effect on the rumen microbial ecosystem with a higher degradation in feed with cellulose such as roughages than those without. In both studies, AFB1 degradability rate was calculated as the difference between initially included AFB1 and residual AFB1 in the culture fluids with no formed metabolites tested for. The remaining AF is absorbed in the small intestines and hydroxylated in the liver to form AFM1, the major metabolite among other metabolites [75], with AFM1 being excreted in milk and urine, and is classified as class 1 carcinogen to humans [22]. The carry-over of AFB1 to milk varies from less than 1% to 6.2% [61,72,76]. The level of carry-over is usually determined by several factors such as the animal species, individual animal variability [72], feeding regimens and type of diet [77], presence of other mycotoxins [32], stage of lactation [76], and actual milk production [76]. AFM2 is also another metabolite of hydroxylation of AFB2 but is of little concern as compared to AFM1. Hernandez–Camarillo et al. [78] reported an occurrence of AFM2 in 20% of cheese in Mexico (mean 0.2 µg/kg) in comparison with 53% reported for AFM1 (mean 3.0 µg/kg) in the same samples [78]. With the higher occurrence of AFB1 in dairy feed in comparison with AFB2 in SSA, AFM1 is therefore of major concern as compared to AFM2. Studies on milk in SSA countries have shown a high incidence of AFM1 (Table 4).

Due to the rumen’s capability to degrade DON, the carry-over of both unmetabolized DON and DOM-1 to milk is negligible. Keese et al. [47] detected no unmetabolized DON in milk using an HPLC–UV method with prior β-glucuronidase incubation in cows fed either 5.3 mg DON/kg dry matter (DM) over 11 weeks, or a ration with 60% concentrate and 4.6 mg/kg DM for 29 weeks. Negligible amounts of DOM-1 (0.21 µg/kg) in two out of 24 samples in the study were detected with LC-MS/MS. In agreement, Seeling et al. [48] using an HPLC–UV method with β-glucuronidase incubation (limit of detection or LOD of 0.5 µg/kg) also reported no unmetabolized DON in milk and DOM-1 at 1.6 and 2.7 µg/kg in cows with 34 mg to 76 mg daily DON intake. Using a more sensitive GC-MS method (LOD of 0.1 µg/kg), DON was detected at levels between 0.1 and 0.3 µg/kg with DOM-1 at levels between 1.5 and 3.1 µg/kg.

Carry-over of FUM to milk is not significant and does not pose a hazard to human health. The occurrence of milk naturally contaminated with FUM does, however, occur, with Maragos and Richard [84] reporting only one sample out of 150 having detectable levels of FB1 using an LC with fluorescence detection method (LOD of 5 ng/mL). Experimentally, Richard et al. [68] did not detect any FUM in milk in two jersey cattle fed 3 mg/kg DMI daily (total 75 mg) for 14 days using two analytical methods (LOD of 5 ng/mL). Similarly, Scott et al. [85] detected no residues of FB1 in the milk of cows dosed with pure FB1 either orally (1.0 and 5.0 mg FB1/kg Body Weight (BW) or by i.v. injection (0.05 and 0.20 mg FB1/kg BW).

Both OTA and its metabolite ochratoxin α can be transferred to milk. Ribelin et al. [69] reported OTA in milk the next day in cows fed on 13.3 mg/kg OTA as a single dose, trace amount of OTA from day 3 to 5 in cows fed 1.66 mg/kg daily for four days, and no OTA in cows dosed with less than 1.66 mg/kg OTA. However, milk from all cows had traces of ochratoxin α. Other experimental studies have reported no carry-over of OTA. Zhang et al. [86] did not detect OTA and ochratoxin α using LC-MS/MS (LOD of 0.1–0.2 ng/mL) in cows administered a single dose of OTA at levels 30 μg/kg OTA BW) in feed. Similarly, Hashimoto et al. [87] detected no OTA in milk of cows fed 100 μg/kg DM for 28 days. Thus, with this negligible rate of carry-over and the low levels of OTA in feed in SSA, OTA poses no health hazard to humans through dairy products.

Little information is available on the levels of these other mycotoxins and their residues in milk in SSA.

### 5.1. East Africa

In East Africa, there is a high prevalence of AFM1 in milk that corresponds to the high levels of AFB1 reported in feeds. Ethiopia, Kenya, and Sudan have reported a 100% incidence of AFM1, with 91.8%, 66.4%, and 100% of the positive samples being above the EU regulatory limit of 0.05 µg/kg [33,80,83]. Imported milk powder in Sudan also had AFM1 at levels between 0.01 and 0.85 µg/kg [83], with 50% exceeding the EU regulatory limit and 33% above the CODEX and EAC regulatory limit of 0.5 µg/kg. Other studies in Kenya have reported AFM1 occurrence in milk between 39.7% and 99% and between 10.4% and 64% exceeded the EU regulatory limit of 0.05 µg/kg [80,81,88,89], with the highest level of 6.9 µg/kg that is way higher than the EU and EAC limit. Similar high AFM1 found in milk samples from Tanzania, with 83.8% of all positive samples exceeding the EU regulatory limit [36].

### 5.2. Central Africa

Despite little data being available of AFB1 in dairy feeds, AFM1 contaminates raw milk and milk products such as cheese and yogurt. A study on the levels of AFM1 in milk and milk products in markets in Burundi and DR Congo showed 100% positive samples with maximum levels of 0.082 and 0.261 µg/kg, respectively. These maximum levels are above the EU regulatory limit [38].

### 5.3. South Africa

High levels of AFM1 occur in milk in South Africa. Dutton et al. [90] reported a 100% incidence of AFM1 in milk from dairy farms, ranging from 0.02 μg/L to 1.5 μg/L. Retail milk was also contaminated with AFM1, at levels of 0.01–3.1 μg/L. Similarly, Mulunda et al. [7] in a study carried out in selected rural areas of Limpopo Province in South Africa reported 100% AFM1 occurrence with 90.6% and 62.1% of the positive samples above South Africa and EU regulatory limit of 0.05 µg/kg in Mapete and Nwanedi area, respectively [7].

### 5.4. West Africa

A high prevalence of AFM1 was found in raw milk and imported milk powder in Nigeria. Oyeyipo et al. [14] reported AF in repacked milk powder in five states in the South West region, Nigeria. Of the milk samples, 53.6% was contaminated with AFM1 but none exceeded the Nigerian regulatory limit of 0.5 µg/kg. However, the maximum level of 0.46 µg/kg was above the EU regulatory limit of 0.05 µg/kg. Interestingly, very high levels of AFB1 above the Nigerian and EU regulatory limit of 5 µg/kg were reported in milk (29.7–79.4 µg/kg) and this can be explained by the frequent presence of *Aspergillus* species that were found contaminating the milk due to the open-air repackaging of the milk powder. In another study on raw milk from free-grazing cows in Abeokuta, Nigeria, Oluwafemi et al. [82] reported a 75% occurrence of AFM1 with 64% exceeding the EU limit.

The high level of AFM1 in SSA is a major food safety concern. A risk assessment by Ahlberg et al. [91] on AFM1 exposure in low- and mid-income dairy consumers in Kenya reported 2.7% of children could hypothetically be stunted due to AFM1 exposure from milk, although stunting has not been proven to occur after exposure to AFM1. Exposure to AFM1 from milk in Kenya has been associated with a reduction in growth, although this is no evidence of causation [92]. However, the hepatocellular cancer risk was low at 0.004 cases per 100,000. In agreement, Sirma et al. [93] reported the annual incidence rates of cancer attributed to the consumption of AFM1 in milk in Kenya between 0.0014 to 0.0039 per 100,000 people. The potential risks to human health have led to several countries setting up legislation for mycotoxins in dairy feeds and milk.

## 6. Mycotoxin Mitigation Strategies

Due to the negative health and economic impact of mycotoxins on the dairy industry and the relative stability of mycotoxins to manufacturing processes, strategies have been developed to mitigate the effects of mycotoxins. Most of these strategies have been developed for control of AF but some are applicable for control of other mycotoxins. There is low awareness on mycotoxins by dairy farmers in SSA with little done in dissemination of information on appropriate control strategies [94]. Kangethe et al. [95] in a study in urban areas in Kenya reported the highest level of awareness on AF at 42%. Similarly, Kirino et al. [81] reported 58% of milk traders being aware of AF but very few had knowledge of AF carry-over to milk [81], and farmers also report feeding moldy maize to animals [23]. In Rwanda, 92.4% of livestock farmers and animal feed vendors were unaware of AF and FUM and their effects [13]. Similarly, Changwa et al. [11] reported a general awareness of mycotoxins between 17% and 92% in South Africa. James et al. [96] reported AF awareness levels of 20.8% among farmers, 26.7% among traders, 60% among poultry farmers, and 25.2% among consumers in Benin, Ghana, and Togo. A low level of awareness has also been reported in Tanzania and Ethiopia [94]. This low level of awareness may hinder the implementation of various mitigation strategies. These strategies are divided into two; pre-harvest strategies that are aimed at preventing the fungal contamination in the field and the post-harvest strategies that are applied to the harvested products during harvesting, processing, or storage to prevent contamination and reduce or eliminate the mycotoxin contamination [97,98]. Prevention of contamination is the preferable method, and post-harvest mitigation strategies are, therefore, very important. However, since this is not always sufficient in SSA, post-contamination options are also needed.

Post-harvest strategies are applied following harvesting. Rapid drying after harvesting reduces the moisture content that is essential in stopping the growth of fungi and mycotoxins production. Moisture content of 10%–13% is considered safe for cereals. However, proper drying and storage is often an issue in most SSA countries due to the high temperatures and humidity [98].

Storage of feed in dry condition with low humidity, proper aeration, and free from rodents and pests is essential for minimizing fungal contamination and mycotoxin production [98].

Most of the dairy feeds are usually bought or harvested and stored, the storage conditions are sometimes unfavorable and hence have a negative effect on feed quality [3].

The tropical climate in SSA is favorable for mycotoxin production by fungi as well as causing issues with food insecurity leading to practices such as diverting moldy grains to be used as animal feeds. This, therefore, makes decontamination the best strategy for the prevention of mycotoxins in the dairy chain [23,99]. Decontamination is applied to the already contaminated feed to eliminate the mycotoxin or to reduce the bioavailability of the toxin [42]. Chemical, physical, and biological methods have been widely applied to decontaminate feed from mycotoxins [5,42,100]. Substances used for decontamination are called detoxifiers and can be grouped into binders that prevent the absorption of mycotoxins and modifiers that break down the mycotoxins in the intestines into less toxic metabolites. Binders usually include clay minerals or yeast products, while modifiers include microorganisms and enzymes [101].

### 6.1. Chemical Decontamination

Chemical methods include the use of acids, bases, aldehydes, bisulphites, oxidizing agents, chlorinating agents, and various gases. These chemicals have been applied and found to be effective against some mycotoxins. Ammoniation has been shown to reduce the levels of OTA in cereals to undetectable levels with these grains being suitable for use in making animal feed without changing the nutritional value [102]. Bailey et al. [103] and Fremy et al. [104] demonstrated the effectiveness of ammonia treatment on the reduction of carry-over of AFM1 to milk in cattle; however, ammoniation is not effective against FUM with contaminated grains still retaining toxicity against rats [105,106]. Sodium bisulphite, hydrogen peroxide, and ozone are also effective in reducing AFB1 contamination in human food [107]. However, these methods are expensive and not easily acceptable by dairy farmers and may affect animal health in vivo due to the accumulation of chemical residues.

### 6.2. Physical Decontamination

Physical methods use adsorbents such as activated charcoal and aluminosilicate clay minerals such as smectite, bentonite, and montmorillonite. These adsorbents act by binding the mycotoxin to prevent its absorption and are effective and safe in ruminants [108,109,110].

Several adsorbents are effective on AF in terms of reducing the carry-over of AFM1 and the effect on animal health. Pietri et al. [109] reported a 41% and 31% decrease in AFM1 in milk in cows fed diet contaminated with 97.3 µg AFB1/kg DMI using 50 g/cow/day and 20 g/cow/day of a commercial detoxifier that contains bentonite, *Eubacterium* and yeast. Kissel et al. [111] in agreement found a 60.4% decrease in AFM1 carry-over in milk in cows fed a diet with 227 g/cow per day of bentonite, and Kutz et al. [60] reported a 45% and 48% decrease in excretion of AFM1 in the milk of cows fed 112 µg AFB1/kg DMI supplemented with two commercial sodium calcium aluminosilicate adsorbents at 0.56% of the diet. In another study, Xiong et al. [112], using sodium montmorillonite with live yeast, yeast culture, mannan oligosaccharide, and vitamin E at a dose of 0.25% of the diet, reported a decrease in the transfer of AFM1 from feed with 20 µg AFB1/kg Dry Matter Intake (DMI) compared with the AFB1 alone control (0.46 vs. 0.56%, respectively). Jiang et al. [57] also reported a reduced transfer of dietary AFB1 as AFM1 in milk and prevention in a decrease in milk yield caused by AFB1. The cows were fed a diet with 75 µg AFB1/kg DMI supplemented with bentonite clay (200 g/cow/day) with or without *Saccharomyces cerevisiae* fermentation product (35 g/cow/day). Sulzberger et al. [58] also further reported a 25% reduction in AF transfer from the rumen to milk using 0.5% clay, 18% (1% clay), and 41% (2% clay) in cows administered 100 µg AFB1/kg DMI. The cows received oral supplementation of 0.5%, 1%, and 2% clay containing vermiculite, nontronite, and montmorillonite. In other ruminants, Mugerwa et al. [110] reported a reduction of AFM1 carry-over in milk in lactating goats fed 100 µg AFB1/kg and 1% DMI activated charcoal and calcium bentonite. None of the adsorbents had any impact on feed intake and milk composition. The efficacy of these adsorbents to bind AF can be affected by the ratio of adsorbent to mycotoxin, the pH, and the temperature. Sumantri et al. [61] reported no effect of inclusion of bentonite at 0.25% and 2% in the diet of cows fed 350 µg AFB1/cow per day for 5 days. Ogunade et al. [56], despite also reporting no reduction in AFM1 concentration after feeding a diet with 75 µg AFB1/kg DMI and 20 g/head/day of a sequestering agent containing sodium bentonite and *S. cerevisiae* fermentation product, reported a reduction of the time required to reduce AFM1 in milk to safe levels after AFB1 contaminated feed withdrawal. However, there is a possibility of some of the adsorbents to bind other nutrients reducing feed nutritive value and feed palatability with EFSA recommending a maximum level of bentonite of 20,000 mg/kg for complete feed [108,113].

Adsorbents have low effectiveness against other mycotoxins. Bhatti et al. [114] in a study on the protective role of bentonite against OTA-induced immunotoxicity in broilers reported partial or no improvement on the negative effects induced by OTA. Binders containing humic acids and mixed-layered smectite-containing binders have been shown to have the capacity to bind to ZEN. De Mil et al. [101] in an in vitro study based on the low level of free ZEN concentration after 4 h incubation with a ZEN: binder ratio of 1:20,000 reported the binding of ZEN.

### 6.3. Biological Decontamination

Biological methods use enzymes or microorganisms to biotransform mycotoxins into less toxic metabolites. BBSH^®^ 797 is a bacterial strain of the family *Coriobacteriaceae* that produces specific enzymes de-epoxidases that open the toxic epoxide ring of trichothecenes, such as DON and T-2 toxin, thus detoxifying them. The yeast strain *Trichosporon mycotoxinivorans* (Trichosporon MTV) is capable of degrading OTA and ZEN [115].

Enzymes used for biotransformation usually cleave the mycotoxin at the site responsible for toxicity in the gastrointestinal tract and produce a metabolite(s) with less toxicity than the parent mycotoxin. AF detoxifizyme (AFDF) from *Armillariella tabascens* [116] and laccase enzyme from *Peniophora* and *Pleurotus ostreatus* fungus [117] have been shown to be AF degrading enzymes. Carboxylesterase and amino-transferase have been shown to degrade FUM [118], with fumonisin esterase being commercially used for decontamination of feed contaminated with FUM [119]. Fumonisin esterase is an FB1 hydrolyzing enzyme that catalyzes the cleavage of FB1′s tricarballylic acid side chains to form partially hydrolyzed FB1a and b (pHFB1a, pHFB1b) and eventually hydrolyzed FB1 (HFB1). This HFB1 is less toxic compared to FB1 [120]. Despite no information being available for their use in dairy cattle, the enzymes are effective in pigs and poultry [119,121].

In Kenya, mycotoxins detoxifiers are used in animal feeds for decontamination. Lack of regulation has led to a lack of information on the efficacy of these detoxifiers and this may expose the farmers to products that are not effective. Mutua et al. [99] in a study on the use of mycotoxin detoxifiers in Kenya showed that usage of binders is uncontrolled, and all nine types of products sold on the Kenyan market were imported as feed additives and not specified as mycotoxin binders. These binders are bought by feed processors and farmers formulating their feed. Information is still lacking on the use and regulation of these products. More information and regulation are, therefore, needed on the use of these detoxifiers in the dairy enterprise in SSA as an alternative strategy of controlling mycotoxins.

### 6.4. Vaccination against AFs

Vaccination as a recently investigated way of reducing AF toxicity, and AFM1 carry-over is a valuable option for AF mitigation in dairy. Polonelli et al. [122] reported up to 46% lower AFM1 levels in milk in vaccinated cows compared to control cows exposed to AFB1. Anaflatoxin B1(AnAFB1) conjugated to keyhole limpet hemocyanin (KLH) together with Freund’s adjuvant was used as a vaccine (AnAFB1-KLH) and produced long-lasting titers of anti-aflatoxin B1 (AFB1) antibodies that also cross-reacted with other AFs. Giovati et al. [123] in an attempt to improve the vaccine used AnAFB1 conjugated to KLH and mixed with complete (priming) and incomplete Freund’s adjuvant (boosters) and reported the average AFM1 concentration in milk collected from vaccinated cows being 74% lower than in milk of control animals. Similarly, the carry-over rate calculated in vaccinated cows (0.77%) was lower than control animals (3.40%). The results show that vaccination can be a feasible option for controlling hazards caused by AFs on animals and humans. However, the production of these vaccines is costly and hence not used as a mitigation strategy.

## 7. Conclusions

It is evident that mycotoxin contamination of dairy feeds is widespread in SSA. Due to low levels of awareness of mycotoxins and food insecurity in SSA, dairy animals will continue to be fed with mycotoxin-contaminated feed. This negatively impacts the dairy industry in SSA due to significant economic losses as a result of the impact on animal health and productivity and food safety due to the contamination of milk. Vital information concerning mycotoxin contamination of dairy feed and products in most African countries, especially Central Africa, is lacking and this may hinder the development of the dairy industry. Furthermore, this information may help in designing effective mycotoxin control strategies in SSA. Mycotoxin detoxifiers may play a significant role in controlling the effects of mycotoxins due to the high levels of contamination of dairy feeds shown in SSA; however, information is lacking on their use and regulation of these detoxifiers, exposing dairy farmers to the risk of using ineffective products. More research and regulation on the use of these detoxifiers may be an effective means of ensuring animal health as well as food safety and security.

## Figures and Tables

**Table 1 toxins-12-00222-t001:** Regulatory and guidance levels of mycotoxins in dairy feed and milk.

Country/Region	Regulatory Limit (µg/kg)			Guidance Values (µg/kg)				
	Total AF	AFB1	Milk AFM1	DON	FUM	OTA	ZEN	Reference
Central Africa region	-	-	-	-	-	-	-	-
East Africa Community	10	5	0.5	-	-	-	-	[16]
West Africa region	-	-	-	-	-	-	-	-
South Africa	10	5	0.05	3000	50,000	-	500	[10]
Rwanda	10	5	-	-	-	-	-	[13]
Nigeria	-	5	0.5	-	-	-	-	[14]
Senegal	-	50	-	-	-	-	-	[12]
Cote d’ Ivorie	10	-	-	-	-	-	-	[12]
Mozambique	10	-	-	-	-	-	-	[12]
CODEX	-	5	0.5	-	-	-	-	[15]
European Union	-	5	0.05	5000	50,000	-	500	[18]
USA	20	-	0.5	-	30,000	-	-	[19]

AF—Aflatoxins, AFB1—Aflatoxin B1, AFM1—Aflatoxin M1, CODEX—Codex Alimentarius DON—Deoxynivalenol, EU—European Union, FUM—Fumonisins, OTA—Ochratoxin A, USA—The United States of America. - Not detected.

**Table 2 toxins-12-00222-t002:** Mycotoxins in dairy feed in Sub-Saharan Africa.

**Aflatoxins**									
**Country**	**Mycotoxin**	**Test**	**Feed**	***n***	**Positive (%)**	**Above EU Limit (%)**	**Max (µg/kg)**	**Mean (µg/kg)**	**Reference**
Ethiopia	AFB1	ELISA	Dairy feed (compounded feed, brewer yeast, silage, maize, and pea hull)	156	100%	100%	419	97	[33]
Ghana	AF	HPLC-FLD	Animal feed and raw materials	18	72%		199	26	[21]
Kenya	AF	ELISA	Animal feed (moldy maize)	207	56%		7.13	3.8	[17]
	AFB1	ELISA	Concentrates and forages	74	57%	56%	147.9	28.3	[6]
	AFB1	ELISA	Compound dairy feed (Manufacturer)	102		62%	4682	9.8	[27]
	AFB1		Dairy feed (Retailers)	31		90%	1198	25.6	
	AFB1		Dairy feed (Farmer)	114		73%	9661	13.7	
	AF	TLC	Dairy feed	72	100%	95%	1123		[34]
	AF	HPLC-FLD	Animal feed and raw materials	27	78%		556	52	[21]
Nigeria	AFB1	HPLC-FLD	Dairy feed	144	87%	66%	24.8	10.5	[35]
	AF	HPLC-FLD	Animal feeds and raw materials	50	94%		435.9	115	[21]
Sudan	AF	HPLC-FLD	Animal feed and raw materials	13	54%		75	90	[21]
South Africa	AFB1	UHPLC-QTOF-MS/MS	Dairy feed	40	48%	62%	3.3	0.7	[11]
	AFB2		Dairy feed	40	93%		23.9	3.1	
	AFG1		Dairy feed	40	55%		19.9	2.6	
	AFG2		Dairy feed	40	100%		116.0	41.3	
	AF	LC-MS/MS	Compounded dairy feeds	25	52%	16%	71.8	14.7	[10]
	AF	HPLC-FLD	Animal feed and raw materials	77	6%		7	0.2	[21]
Tanzania	AFB1	ELISA	Spoilt maize	41	29%			3.5	[28]
	AFB1		Maize bran	20	60%			3.3	
	AFB1	HPLC	Sunflower based dairy feed	20	65%	62%	20.5		[36]
**Type B Trichothecenes**									
**Country**	**Mycotoxin**	**Test**	**Feed**	***n***	**Positive (%)**		**Max (µg/kg)**	**Mean (µg/kg)**	**Reference**
Ghana	Type B trichothecenes (DON, 3/15-Ac-DON, and NIV)	HPLC-UV	Animal feed and raw materials	18	50%		1550	955	[21]
Kenya	Type B trichothecenes (DON, 3/15-Ac-DON, and NIV)	HPLC-UV	Animal feed and raw materials	25	48		3859	422	
	Type B trichothecenes (DON, 3/15-Ac-DON, and NIV)	ELISA	Concentrates and forages	74	63%		180	49	[6]
Nigeria	Type B trichothecenes (DON, 3/15-Ac-DON, and NIV)	HPLC-UV	Animal feeds and raw materials	45	58%		463	316	[21]
Sudan	Type B trichothecenes (DON, 3/15-Ac-DON, and NIV)	HPLC-UV	Animal feed and raw materials	9	33%		353	100	[21]
South Africa	Type B trichothecenes (DON, 3/15-Ac-DON, and NIV)	HPLC-UV	Animal feed and raw materials	77	87%		11,022	1469	[21]
	DON	UHPLC-QTOF-MS/MS	Dairy feed	40	60%		82	20	[11]
	DON	LC-MS/MS	Compounded dairy feeds	25	96%		2280	891	[10]
**Fumonisins**									
**Country**	**Mycotoxin**	**Test**	**Feed**	***n***	**Positive (%)**		**Max (µg/kg)**	**Mean (µg/kg)**	**Reference**
Ghana	FUM	LC-MS	Animal feed and raw materials	18	89%		929	500	[21]
Kenya	FUM	LC-MS	Animal feed and raw materials	25	76		10,485	956	[21]
Nigeria	FUM	LC-MS	Animal feeds and raw materials	45	78%		2860	919	[21]
Sudan	FUM	LC-MS	Animal feed and raw materials	9	11%		23	23	[21]
South Africa	FUM	LC-MS/MS	Compounded dairy feeds	25	100%		2497	975	[10]
	FB1	UHPLC-QTOF-MS/MS	Dairy feed	40	85%		1390	373	[11]
	FUM	LC-MS	Animal feed and raw materials	77	57%		4398	454	[21]
Tanzania	FUM	ELISA	Spoilt maize	41	51%			14,450	[28]
	FUM	ELISA	Maize bran	20	60%			1630	
**HT-2 Toxin**									
**Country**	**Mycotoxin**	**Test**	**Feed**	***n***	**Positive (%)**		**Max (µg/kg)**	**Mean (µg/kg)**	**Reference**
South Africa	HT-2	UHPLC-QTOF-MS/MS	Dairy feed	40	88%		313	35	[11]
**Ochratoxin a**									
**Country**	**Mycotoxin**	**Test**	**Feed**	***n***	**Positive (%)**		**Max (µg/Kg)**	**Mean (µg/Kg)**	**Reference**
Kenya	OTA	HPLC-FLD	Animal feed and raw materials	2	50%		2	2	[21]
Nigeria	OTA	HPLC-FLD	Animal feeds and raw materials	5	100%		12	12	[21]
Sudan	OTA	HPLC-FLD	Animal feed and raw materials	6	67%		19	15	[21]
South Africa	OTA	LC-MS/MS	Compounded dairy feeds	25	16%		17	10	[10]
**Zearalenone**									
**Country**	**Mycotoxin**	**Test**	**Feed**	***n***	**Positive (%)**		**Max (µg/kg)**	**Mean (µg/kg)**	**Reference**
Ghana	ZEN	HPLC-FLD	Animal feed and raw materials	18	11%		310	178	[21]
Kenya	ZEN	HPLC-FLD	Animal feed and raw materials	25	56%		167	67	[21]
Nigeria	ZEN	HPLC-FLD	Animal feeds and raw materials	45	51%		80	46	[21]
South Africa	ZEN	UHPLC-QTOF-MS/MS	Dairy feed	40	60%		28	3	[11]
	ZEN	LC-MS/MS	Compounded dairy feeds	25	96%		123	72	[10]
	ZEN	HPLC-FLD	Animal feed and raw materials	77	29%		195	86	[21]

AF—Aflatoxins, AFB1—Aflatoxin B1, AFB2—Aflatoxin B2, AFG1—Aflatoxin G1, AFG2—Aflatoxin G2, Ac-DON—3/15-Acetyl-deoxynivalenol, DON—Deoxynivalenol, ELISA—Enzyme-linked Immunosorbent Assay, FB1—Fumonisin B1, FUM—Total Fumonisins, HPLC-UV—High-Performance Liquid Chromatography with Ultraviolet Detection, HPLC-FLD—High-Performance Liquid Chromatography with Fluorescent Detection MEAN-Mean of positives, LC-MS/MS—Liquid Chromatography Tandem Mass Spectrometry, NIV—Nivalenol, UHPLC-QTOF-MS/MS—Ultra High-performance Liquid Chromatography coupled with Quadrupole Time of Flight tandem Mass Spectrometry, ZEN—Zearalenone.

**Table 3 toxins-12-00222-t003:** Effect of mycotoxins in dairy cattle.

Effect	AF	DON	FUM	OTA	T-2	ZEN
Reduced feed intake	√	√	√	√	√	√
Reduced milk yield	√	√	√	√	√	√
Reproductive effects	√		√		√	√
Immunosuppression	√				√	
Hepatotoxicity	√		√			
Nephrotoxicity	√		√			
Gastroenteritis				√	√	

AF—Aflatoxin, DON—Deoxynivalenol, FUM—Fumonisin, OTA—Ochratoxin A, T-2—T-2 toxin, ZEN—Zearalenone. √ Effect present.

**Table 4 toxins-12-00222-t004:** Aflatoxin M1 in milk in Sub-Saharan Africa.

Country	Test	Sample	*n*	Positive (%)	Above Eu Limit (%)	Max (µg/Kg)	Mean (µg/Kg)	Reference
Burundi	ELISA	Milk (fresh and yoghurt)	16	100%		0.08	0.03	[38]
D.R. Congo	ELISA	Milk (fresh and yogurt) and cheese	10	100%		0.26	0.03	[38]
Ethiopia	ELISA	Milk	110	100%	91.8%	4.98	0.4	[33]
Kenya	ELISA	Milk	96	100%	66.4%	4.63	0.29	[79]
	ELISA	Milk	291		51.9%	1.1	0.08	[80]
	ELISA	Milk	512	39.7%	10.4%	6.9	0.003	[27]
	ELISA	Milk	200		55%	1.67	0.128	[81]
Nigeria	HPLC	Milk powder	125	53.6%		0.46		[14]
	HPLC	Raw milk	100	75%	64%	0.46	0.11	[82]
Sudan	Fluorometry	Raw milk	35	100%	100%	2.52	0.92	[83]
		Imported powder milk	12			0.85	0.29	
South Africa	ELISA	Milk	30	100%	90.6%	0.15	0.09	[7]
		Milk	37	100%	62.1%	0.11	0.07	
Tanzania	HPLC	Milk	37	83.8%	100	2.01		[36]

ELISA—Enzyme-Linked Immunosorbent Assay. HPLC—High-Performance Liquid Chromatography, EU Limit—0.05 µg/kg.

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
