# Peer review of "A Review of the Impact of Mycotoxins on Dairy Cattle Health: Challenges for Food Safety and Dairy Production in Sub-Saharan Africa"

_toxins, 2020, doi:10.3390/toxins12040222_

Round 1
Reviewer 1 Report
This review is both timely and comprehensive and I believe it would provide a significant resource for the readership of the journal.
Page 1 line 19 - the review only covers post-harvest strategies and thus I would recommend adding post-harvest between up-to-date and mitigation
Page 2 line 58 - the term should be clinical signs and not symptoms.
Page 3 Table 1. Check the value for CODEX for AFM1. I believe CODEX uses 0.5 ppb
Page 4 line 154-155 - the term 'bad sorted" should be better described it is difficult to understand what is meant. I understand that the journal may have used the term but the review would benefit from this description.
Page 1 (6) line 249 - Should be more passive. "In comparison, ruminants can be less severely affected...which is attributed to microbial activity" The information presented clearly shows that in some cases the microbes can amplify toxicity. The use of 'are' is definite.
Page 2 (7) line 279 - Protozoa play a critical role in ruminal degradation and it would add value to state that things like diet and additives (i.e. rumensin/monensin) can alter protozoa populations significantly and thus drastically changing toxin metabolism potential.
Page 4 (9) line 366-373 - It is clear from the review that of the mycotoxins studied only Aflatoxin is a risk for milk carryover. Therefore I would recommend the authors reword the opening statement/paragraph to clarify this point. something to the effect of "of all the studied mycotoxins it is the only aflatoxin that has been described to transfer to milk of lactating cattle. This is of great importance to public health as the toxin is classified as a carcinogen and infant, the primary consumers of milk, are susceptible" I am concern that the open statements on this first paragraph incorrectly paint a picture of a bigger problem than is described further on. It is, however, pertinent to comment that the is the risk associated with emerging toxins.
Page 7 (12) line 484 - the authors should clarify here that only post-harvest methods will be covered in this review. The paragraph leaves the impression that both pre and post-harvest methods will be included.
Page 7 (12) line 528 and others - I would caution the authors on the use of brand names in this review. I would at least make sure to make a statement that products evolve and may change sources of raw materials over the evolution of product development and thus impact their performance.
References: I advise authors to check reference formating for consistency. Some references have bolded years some don't. Also, some references have formatting errors at the beginning like the number inside parenthesis (i.e. line 836) or a tab (line 959).
Reviewer 2 Report
Line 80, 324 - there must be a gap between the number and the unit.
Table 1,2,3 and 4 - edit the headings in the columns,
- adjust the column width,
- use abbreviations for mycotoxin names (Table 3).
Line 100 - missing a comma after the word Union.
Line 137 - erroneously dot after sentence.
Line 208 - missing gap between the reference and bracket.
Author Response
REVIEWER TWO RESPONSE
Line 80, 324 - there must be a gap between the number and the unit.
Response 1: Done
Table 1,2,3 and 4 - edit the headings in the columns,
- adjust the column width,
- use abbreviations for mycotoxin names (Table 3).
Response 2: Heading edited for table 1, 2, 3 and 4 and abbreviations used for table 3
Line 100 - missing a comma after the word Union.
Response 3: Line 102 - Done
Line 137 - erroneously dot after sentence.
Response 4: Corrected
Line 208 - missing gap between the reference and bracket.
Response 5: Corrected
Reviewer 3 Report
This in an interesting and extent study presenting a review of mycotoxins effects and impacts on dairy cattle from several aspects. Although paper is interesting and related with Journals' scope there are some points preventing me for accepting paper in its current format.
Please find bellow some points to be considered for corrections:
1.Page 1 line 42: Please define the terms of poor quality (i.e. fungus infections? high moisture?) since quality include several aspects.
2.Page 2 line 48-50. Production of mycotoxins where? stress and damage caused to whom?
3.Page 3. line 114-115. "bad maize" better rephrase this term since its not clear to what "bad" corresponds to. On top of this its better to use more scientific terminology
4.Page 4 line 122-123 high toxicity to whom? (animals, humans?) frequent occurrence where? (feeds, milk, humans?). Also its not clear what authors want to say with the following phrase "....as well as their regulations in most countries"
5.Structure for presenting results in East Africa, Central Africa, West Africa etc is not the more appropriate. Better delete those Sub Titles and mentioned this at the beginning of the first paragraph i.e For Central Africa ...... or make them real subtitles i.e 3.1 East Africa.
6.Table 2. page 1 of the table. There is no number for GHANA Animal feed and raw materials correspond to ref. 21 and no percentage of positive samples is given although there are levels of afl above EU limit
7.Chapter 4. line 224-225 better rephrase since food security in not the most appropriate term.
8.Chapter 4. Aflatoxins, DON, Fumonisins, OTA etc should be presented as described in point 5
Round 2
Reviewer 3 Report
All suggested corrections have been applied. Paper could be accepted under current format.